

# Macromolecular composition of terrestrial and marine organic matter in sediments across the East Siberian Arctic Shelf

Robert B. Sparkes[1,2], Ayça Doğrul Selver[1,3], Örjan Gustafsson[4], Igor P. Semiletov[5,6,7],
Negar Haghipour[8], Lukas Wacker[8], Timothy I. Eglinton[8], Helen M. Talbot[9], and Bart E. van Dongen[1]

[1]School of Earth, Atmospheric and Environmental Sciences and Williamson Research Centre for Molecular Environmental Science, University of Manchester, UK
[2]School of Science and the Environment, Manchester Metropolitan University, Manchester, UK
[3]Balıkesir University, Geological Engineering Department, Balıkesir, Turkey
[4]Department of Environmental Science and Analytical Chemistry (ACES) and the Bolin Centre for Climate Research, Stockholm University, Sweden
[5]Pacific Oceanological Institute Far Eastern Branch of the Russian Academy of Sciences, Russia
[6]International Arctic Research Center, University of Alaska, USA
[7]National Tomsk Research Polytechnic University, Russia
[8]Laboratory of Ion Physics, ETH, Honggerberg, Zurich, Switzerland
[9]School of Civil Engineering and Geosciences, Newcastle University, UK

*Correspondence to:* Robert Sparkes (r.sparkes@mmu.ac.uk)

**Abstract.** Mobilisation of terrestrial organic carbon (terrOC) from permafrost environments in Eastern Siberia has the potential to deliver significant amounts of carbon to the Arctic Ocean, via both fluvial and coastal erosion. Eroded terrOC can be degraded during offshore transport, or deposited across the wide East Siberian Arctic Shelf (ESAS). Most studies of terrOC on the ESAS have concentrated on solvent-extractable organic matter, but this represents only a small proportion of the total

5 terrOC load. In this study we have used pyrolysis gas chromatography mass spectrometry (py-GCMS) to study all major groups of macromolecular components of the terrOC; this is the first time that this technique has been applied to the ESAS. This has shown that there is a strong offshore trend from terrestrial Phenols, Aromatics, Cyclopentenones to marine Pyridines. There is good agreement between proportion Phenols measured using py-GCMS and independent quantification of lignin phenol concentrations ($r^2$ = 0.67, p < 0.01, n = 24). Furfurals, thought to represent carbohydrates, show no offshore trend and are

10 likely found in both marine and terrestrial organic matter. We have also collected new radiocarbon data for bulk OC ($^{14}C_{OC}$) which, when coupled with previous measurements, allows us to produce the most comprehensive $^{14}C_{OC}$ map of the ESAS to date. Combining the $^{14}C_{OC}$ and py-GCMS data suggests that the Aromatics group of compounds is likely sourced from old, aged terrOC in contrast to the Phenols group, which is likely sourced from modern woody material. We propose that an index of the relative proportions of Phenols and Pyridines can be used as a novel terrestrial *vs.* marine proxy measurement for

15 macromolecular organic matter. Principal component analysis found that various terrestrial *vs.* marine proxies show different patterns across the ESAS, and shows that multiple river-ocean transects of surface sediments transition from river-dominated to coastal erosion-dominated to marine-dominated signatures.



# 1 Introduction

Northern hemisphere permafrost is a significant and vulnerable store of organic carbon (OC), containing approximately 40% of the global soil OC budget (northern hemisphere terrestrial permafrost contains at least 1330 – 1580 Gt OC, other biomes contain 2050 Gt OC; Schuur et al., 2015). The vast amount of soil OC currently freeze-locked in the permafrost is vulnerable

to global warming and can be remobilised through permafrost thawing, increased river runoff and coastal erosion (Stendel and Christensen, 2002; Vonk et al., 2010, 2015). Recent studies show that the Arctic region is warming twice as fast if compared to other parts of the world (IPCC, 2013) and that both the flux and nature of remobilised terrOC is projected to change in the coming decades (Holmes et al., 2002, 2012; van Dongen et al., 2008a; O'Donnell et al., 2014). Indeed, in parts of the Eurasian Arctic region, global warming caused an increase in permafrost temperatures of up to 2 °C between 1971 – 2010

(Schuur et al., 2008) and up to 7% increase in discharge rates of the main Eurasian rivers (Peterson et al., 2002). Coupled warming and increased discharge is releasing 'old' carbon from thawing permafrost, previously stored for thousands of years (Gustafsson et al., 2011; Feng et al., 2013, 2015b; Vonk et al., 2012), via active layer deepening and thermokast erosion events (Vonk and Gustafsson, 2013; IPCC, 2013). Additionally, coastal erosion is an important process through which vast amounts of terrOC are transported to the Arctic shelf (Vonk et al., 2012). In particular, the East Siberian coastline is dominated by Ice

Complex Deposits (ICD; also known as "Yedoma"). These are Plio-Pleistocene permafrost deposits rich in OC, deposited in steppe-tundra environments (Lantuit et al., 2013; Schirrmeister et al., 2008, 2011; Strauss et al., 2012, 2013; Vonk et al., 2010, 2012). Coastal erosion transports $44 \pm 10$ Mt of terrOC to the East Siberian Arctic Shelf (ESAS) annually (Vonk et al., 2012). This amount will likely increase in the next decades due to diminishing sea ice cover resulting in increased storm frequency and wave fetch (Stein and MacDonald, 2004; Vonk et al., 2010).

The fate of terrOC in the Arctic region is still a matter of debate. Susceptibility to degradation will depend on the molecular composition of the terrOC being released, the chemical conditions present in the water column and surface sediments, and the physical characteristics of transport (time spent in water column, sediment transport style, turbidity). Stein and MacDonald (2004) assumed that degradation rates of terrestrial particulate organic carbon (POC) in coastal Arctic environments were comparable to the global average degradation rate of riverine POC. This suggests that a substantial amount of the terrOC is

not degraded during transport across the Arctic shelves but is preserved in marine sediments or delivered to the deep ocean. However, recent studies suggested that a much greater proportion of the river transported terrOC in this region is degraded in the water column, mainly close to the point of origin (Karlsson et al., 2011; Sánchez-García et al., 2011; Semiletov et al., 2007, 2012; Tesi et al., 2014, 2016; van Dongen et al., 2008b; Vonk et al., 2012). van Dongen et al. (2008b), for instance, showed that 65% of terrestrial POC transported by the sub-Arctic Kalix River is degraded in the inner low salinity zone (within

60 km of the river). Ice Complex material has also been shown to be labile upon its remobilisation (Tesi et al., 2016; Vonk et al., 2013a, b; Zimov et al., 2006). Tesi et al. (2016) showed that different components of the carbon load, in both fluvial and coastal erosion sediments, will deposit and degrade at different rates, and therefore terrOC can be considered to exhibit non-uniform behaviour. Density and particle size separations found that OC in topsoil and ICD sediments was distributed between large, low-density particles of plant matter and high-density fine particles (<38 μm). However, the large, low-density



particles were rapidly deposited in nearshore sediments and in the distal ESAS the OC was predominantly found in the fine and ultrafine high-density particles, suggesting the presence of mineral-OC complexes. They showed that some biomarker molecules could exhibit up to 98% degradation across the shelf, and produced average degradation rates of up to 90% for OC in topsoil and 60% for OC from Ice Complexes. This indicated that that patterns seen offshore the ESAS are a combination of
hydrodynamic sorting of OC-bearing particles and degradation of terrOC during cross-shelf transport. Overall, these studies of terrOC transport and degradation suggest that the degradation extent used by Stein and MacDonald (2004) is likely an underestimate and that a much greater proportion of the remobilized terrOC will be degraded and released into the atmosphere as greenhouse gases than previously thought. This will lead to a positive feedback with global climate warming, with the greenhouse gas release translocated from the point of original thaw (Anderson et al., 2009; Alling et al., 2010; Sánchez-García
et al., 2011; Vonk and Gustafsson, 2013; IPCC, 2013). Furthermore, before being vented from the surface ocean as $CO_2$, this degraded terrOC will cause severe ocean acidification of the ESAS (Semiletov et al., 2016).

Previous studies looking into the composition and fate of terrOC transported to the Arctic shelf primarily focused largely on the extractable fraction of the terrOC (that which can be isolated using a combination of organic solvents; Belicka and Harvey, 2009; Bischoff et al., 2016; Doğrul Selver et al., 2012, 2015; Drenzek et al., 2007; Fernandes and Sicre, 2000; Gustafsson
et al., 2011; Karlsson et al., 2011; Sánchez-García et al., 2014; Sparkes et al., 2015; van Dongen et al., 2008b, a; Vonk et al., 2010, 2012; Yunker et al., 1995). Much less is known about the non-extractable portion. This non-extractable OC constitutes the largest proportion of bulk OC transported to the Arctic Ocean and contains macromolecular components such as lignin, proteins and cellulose, and the degradation products of these. Only a few studies have analysed macromolecular terrOC transported to the Eurasian Arctic shelves, and they have only sampled a small fraction of the shelf area (Feng et al., 2013; Guo et al., 2004;
Peulvé et al., 1996; Tesi et al., 2014; Winterfeld et al., 2015). Guo et al. (2004), for instance, analysed the macromolecular OC compositions of Great Russian Arctic Rivers (GRARs) estuary surface sediments using pyrolysis-gas chromatography mass spectrometry (py-GCMS). Based on the increasing relative abundance of carbohydrate moieties towards eastern Siberia, they suggested that terrOC transported to the Arctic Ocean via the eastern GRARs (from estuaries dominated by continues permafrost) was less degraded compared to OC transported by the western GRARs (from estuaries no longer dominated by
continues permafrost). Feng et al. (2013) showed, by analysing the radiocarbon age of specific lignin moieties in the same set of GRAR sediments, that the vascular plant derived lignin phenols may have originated from young surface soils but that wax lipids mainly originated from deeper permafrost horizons, implying that climate warming may cause old permafrost carbon remobilisation. Recent work investigating macromolecular moieties across the Arctic found a large concentration of plant-derived compounds on the ESAS (Feng et al., 2015a, b), ascribed to either enhanced preservation of these in the exceptionally
cold climate, increased ICD input (Vonk et al., 2012) or a lack of marine and bedrock-derived OC in the area (Semiletov et al., 2005).

It also remains poorly understood how macromolecular terrOC behaves after it is transported to the Arctic Ocean. Recent analyses on the ESAS indicate that lignin may degrade faster than wax lipids (Tesi et al., 2014), suggesting that macro-molecular terrOC may also behave non-uniformly. However, lignin represents only one part of the macromolecular fraction
of the remobilized terrOC so it remains unclear to what extent these results are representative for the entire macromolecular





fraction. Analysis by py-GCMS is a rapid method of investigating macromolecular OC (Guo et al., 2004, 2009; Xu et al., 2009). Flash-heating in an oxygen-free atmosphere produces thermal breakdown products which are GC-amenable, however the thermal breakdown caused by the pyrolysis process can produce hundreds or thousands of different compounds, leading to complex ion chromatograms. This study uses a modified approach in which a small number of dominant compounds are used to represent groups of moieties (Guo et al., 2004, 2009). We aim to use this approach to better understand the origin and fate of macromolecular organic matter on the ESAS. We will demonstrate this innovative technique in the (relatively) well-constrained Kolyma River outflow system, apply it to the entire ESAS and compare it to other macromolecular OC methods to demonstrate that py-GCMS is a rapid and robust procedure for macromolecular OC characterisation. We will then use py-GCMS to differentiate between various terrestrial and marine sources of OC in Arctic Permafrost environments, and to study, for the first time ever, their-cross shelf distributions.

## 2 Method

### 2.1 Study area and sample collection

Samples used in this study were collected from the East Siberian Arctic Shelf. This is a region extending from 130 to 175 °E and from 70 to 77 °N, fed by four of the major GRARs (from west to east: Lena, Yana, Indigirka and Kolyma; see Figure 1). Onshore, the East Siberian Arctic region consists largely of continuously permafrosted land, where soil temperatures remain below 0 °C year-round and are impenetrable to water. The uppermost soils ("active layer") have an annual freeze-thaw cycle and support tundra and taiga vegetation. Enhanced climate warming in the next century is expected to deepen the active layer and increase the permeability of the permafrost layer (Feng et al., 2015b). This process will likely lead to the mobilisation of OC from deeper, older permafrost horizons, a process already observed in north-western Russia and Scandinavia (Feng et al., 2013).

Water depth across the ESAS is very shallow, less than 100 m for several hundred km, before dropping steeply at the shelf break. The ESAS seafloor consists largely of buried permafrost that was inundated during the Holocene (Kienast et al., 2005). The major GRARs (Lena, Yana, Indigirka and Kolyma) deliver 523, 32, 54 and 122 $\mathrm{km}^3\,\mathrm{y}^{-1}$ of water and 21, 4, 11 and 10 Mt sediment respectively each year (Gordeev, 2006). In addition to fluvial input, coastal erosion also plays an important role in sediment and OC delivery. Coastal erosion rates in the ESAS region are among the fastest in the Arctic, measuring up to 10 $\mathrm{m\,y}^{-1}$ (Lantuit et al., 2011). Erosion rates from ICD are 5 – 7 times greater than other coastal permafrost (Vonk et al., 2012, and references therein). Sections of particularly rapid coastal erosion regions are highlighted on Figure 1. Vonk et al. (2012) used dual $\delta^{13}\mathrm{C}$-$\Delta^{14}\mathrm{C}$ isotope measurements of bulk sedimentary OC on the ESAS to show that erosion of coastal ICD was responsible for a large proportion of the sediment and OC input to the ESAS. Biomarker investigations based on extracted sediments have been able to identify and model the contribution from fluvial and coastal delivery processes separately (Bischoff et al., 2016; Doğrul Selver et al., 2015; Sparkes et al., 2015). Tesi et al. (2016) showed that biomarker and radiocarbon values differed between areas dominated by ICD (from coastal erosion) and topsoil (from river erosion), and that these values varied between size and density fractions. The annual OC delivery into this area is estimated to be 10 $\mathrm{MtC\,y}^{-1}$ (Rachold et al.,



2004), roughly one third of the OC delivered annually by the Amazon River (36 $\mathrm{MtC\,y^{-1}}$; Richey et al., 1990), whilst Vonk et al. (2012) estimated 44 $\mathrm{MtC\,y^{-1}}$ from ICD. Differences have been observed between regions of the ESAS using a number of geochemical measurements. Semiletov et al. (2005) found that the western part of the nearshore East Siberian Sea was dominated by freshwater flux and coastal erosion, whilst the eastern part was influenced by water from the Pacific Ocean

entering the region through the Bering Strait. The transition zone between these two regions is variable depending on ocean currents, but is approximately 160 °E, east of the Kolyma River outflow (Semiletov et al., 2005). Tesi et al. (2014) studied CuO-derived lignin moieties on the ESAS and showed that the eastern and northern parts of the shelf were rich in marine OC tracers (e.g. high C16:1 fatty acids) and lower in terrestrial markers (lignin and cutin products). Karlsson et al. (2015) showed that while terrOC was present across the ESAS, organic matter degradation in the eastern region, offshore the Kolyma River,

was dominated by degrading marine OC, whilst in the western areas degradation was typically of terrOC.

Based on these findings, this study has sub-divided the ESAS into four smaller areas (see Figure 1). The "Nearshore Laptev Sea" zone (NLS) contains samples close to the eastern outflows of the Lena River delta. This includes the Buor-Khaya Bay between the Lena Delta and Cape Buor-Khaya. Suspended material and surface sediments are rich in terrOC (Charkin et al., 2011; Karlsson et al., 2011; Winterfeld et al., 2015). GDGT biomarkers in this area are dominated by river-derived material

(Sparkes et al., 2015), but there are also areas of rapid coastal erosion (Muostakh Island, Cape Buor-Khaya) which deliver large amounts of sediment and OC to the bay. However, these coastal erosion sediments have noticeably different isotopic and biomarker signatures when compared to river sediments (Bischoff et al., 2016; Vonk et al., 2012).

To the East of the NLS is the "Dmitry Laptev Strait" zone (DLS). This area, between the mainland and the New Siberian Islands, is situated next to a rapidly eroding coastline (up to 10 $\mathrm{m\,y^{-1}}$; Lantuit et al., 2011) but is over 300 $\mathrm{km}$ from major river

inputs. Both bulk measurements ($\delta^{13}\mathrm{C}_{OC}$ and the Branched and Isoprenoidal Tetraether index (BIT; based on glycerol dialkyl glycerol tetraether lipids) in this area show very strongly terrestrial values (Sparkes et al., 2015; Vonk et al., 2012) – mainly due to a relatively low input of marine OC in this area.

The "Nearshore East Siberian Sea" (NESS) zone covers samples up to 200 $\mathrm{km}$ from the Indigirka and Kolyma river mouths. This region contains the terrOC-dominated section of the East Siberian Sea, as determined by Semiletov et al. (2005) and shown

in data from Tesi et al. (2014). This area is affected by influx from the Indigirka and Kolyma Rivers, as well as the Oyagosski Yar region of extensive coastal erosion to the west of the Indigirka and between the two rivers (Lantuit et al., 2011, Figure 1). The Kolyma River outflow has been extensively studied as a terrestrial-marine transect (Doğrul Selver et al., 2015; Karlsson et al., 2015; Vonk et al., 2010), and disparities have been found between various organic geochemical proxies for terrestrial vs. marine organic matter. Whilst bulk stable carbon isotopes and the bacteriohopanepolyol-based R'soil proxy (Doğrul Selver

et al., 2012) show linear trends offshore, the BIT index decreases rapidly in the first 0 – 200 $\mathrm{akm}$ offshore, leading to a non-linear correlation between the terrestrial vs. marine proxies (Doğrul Selver et al., 2015). This finding can be explained by varying contributions to the bulk OC signal from different OC sources (Sparkes et al., 2015) and settling-fractionated sediment sorting (Tesi et al., 2016).

The "Offshore Arctic Shelf" (OAS) zone contains offshore sections of the Laptev and East Siberian seas, further than 200 $\mathrm{km}$

from the mouths of the Great Russian Arctic Rivers. This region stretches from offshore the Lena River to offshore the Kolyma



River. Bulk isotopic measurements and terrestrial-marine biomarker proxies from these samples all show lower amounts of terrestrial OC and a dominance of marine OC in this area (Bischoff et al., 2016; Doğrul Selver et al., 2015; Sparkes et al., 2015; Tesi et al., 2016; Vonk et al., 2012).

36 surface sediment samples from across the ESAS were used in this study (Figure 1), along with six ICD samples from two terrestrial sample sites. The offshore surface sediments were collected during the International Siberian Shelf Study research cruise in 2008 (ISSS-08; Semiletov and Gustafsson, 2009) by a GEMAX dual gravity corer or a van Veen grab sampler. Sediment cores were sliced into 1 cm sections and transferred into pre-cleaned polyethylene containers, grab samples were sub-sampled using stainless steel instruments into pre-cleaned polyethylene containers. ICD samples were collected from the upper, middle and lower portions of river-bank profiles. All samples were kept frozen before stabilisation by freeze or oven drying (50 °C). The sample sediments used for py-GCMS analysis were previously solvent extracted for biomarker analysis (Sparkes et al., 2015). Briefly, an ultrasonic extraction process using methanol, dichloromethane and pH-buffered distilled water was used to remove extractable material, representing approximately 5% of the total OC content. The sample residues were dried and stored at room temperature prior to analysis in this study. Thus sample names used in this study are equivalent to those reported in Sparkes et al. (2015) as well as other papers based on the ISSS-08 cruise (Karlsson et al., 2015; Tesi et al., 2014; Vonk et al., 2012).

## 2.2 $^{14}C_{OC}$ measurements

In addition to existing radiocarbon data (Vonk et al., 2012), bulk radiocarbon measurements were carried out at the Accelerator Mass Spectrometer (AMS) facility of the Laboratory of Ion Beam Physics (LIP) of the Swiss Federal Institute of Technology (ETH-Zurich, Switzerland). Samples were fumigated in 8x8x15 mm silver boats (Elemental) with HCl >37% under vacuum in a desiccator (Komada et al., 2008), followed by neutralisation for at least 24 hours with NaOH. Prior to Elemental Analysis (EA) combustion, the samples were wrapped in a tin boat 8x8x15mm (Elemental). Samples were measured in gas form with an EA directly coupled to the AMS. Gas targets were measured using the MICADAS instrument at the AMS facility of the Laboratory of Ion Beam Physics (LIP) At ETH-Zurich. Samples have been corrected against an in-house anthracite coal blank and oxalic acid II standard reference material (NIST SRM 4990C).

## 2.3 Pyrolysis-Gas Chromatography-Mass Spectrometry

Dried solvent-extracted residues were analysed using py-GCMS. All samples were analysed using an Agilent GC/MSD system interfaced to a CDS-5200 pyroprobe. Briefly, 10-15 mg of sediment was placed into a clean fire-polished quartz tube along with a known amount of internal standard (5α-androstane) and pyrolysed at 700 °C for 20 s in a flow of Helium. The resulting material was transferred via a heated transfer line to an Agilent 7980A gas chromatograph (GC) fitted with an Agilent HP-5 column coupled to an Agilent 5975 MSD single quadrupole mass spectrometer in electron ionisation mode (scanning a range of m/z 50 to 650 at 2.7 scans s$^{-1}$; ionisation energy 70 eV). The pyrolysis transfer line and rotor oven temperatures were set at 325 °C, the heated GC interface at 280 °C, the EI source at 230 °C and the MS quadrupole at 150 °C. Helium was used as the carrier gas and the samples were introduced in split mode (split ratio 20:1, constant flow of 20 ml min$^{-1}$, gas saver mode





active). The oven was programmed from 40 °C (held for 5 min) to 250 °C at 4 °C $\mathrm{min}^{-1}$, before being heated to 300 °C at 20 °C $\mathrm{min}^{-1}$ and held at this temperature for 1 min for a total run time of 61 minutes per sample. Each sample was run at least in triplicate.

## 2.4 Typical macromolecular moieties used as representative compounds

Py-GCMS produces complex chromatograms containing hundreds of compounds. Approximately 70 of the most abundant pyrolysis moieties were identified (Figure S1 and Table S1). Compounds were identified by comparison of relative retention times and spectra to those reported in the NIST library. Given the complexity of the GCMS chromatograms (see Figure S1) it was not possible to integrate individual compounds in total ion current mode due to significant overlap between ion peaks. Instead, single ion filtering was used to measure the peak area of each compound. The major ion of each compound was filtered and integrated. In line with the approach taken in Guo et al. (2009) a selection of nine representative moieties were chosen that represent key compound classes, many of which can be linked to particular groups of terrestrial or marine macromolecular materials. For example, phenol is a key component of lignin, so can potentially be used as a proxy for the pyrolysis products of terrestrial plant material, although it is also found in other compounds including tannins. Pyridine, a nitrogen-containing aromatic compound, is likely sourced from proteins, which can be found in soils but will mostly come from marine primary productivity in offshore samples. Representative compounds and their inferred sources can be found in Table 1. These compounds are identical to those analysed by Guo et al. (2009) except for the addition of pyridine and methyl pyridine in the "Pyridines" group for the present study. Following the "abundance index" approach of Guo et al. (2004, 2009), the relative areas of the major ions in each group were compared to the total area of all measured compounds and are reported in Table S2. As discussed by Guo et al. (2004, 2009), this approach does not attempt to represent all organic compounds in the sediments, and the relative areas of major ions does not correspond to the actual abundance of each compound. However, this approach uses the most important compounds to demonstrate differences between sediment samples in a defined manner. Expanding on the work of Guo et al. (2004), this study includes samples from across the shelf, rather than just river mouths. Thus the E-W transect can be extended to a whole-shelf survey of macromolecular OC, and the spatial resolution increased.

## 3 Results

### 3.1 Bulk radiocarbon measurements

Radiocarbon values ranged from -748 ‰ to -313 ‰ (see Table S3). The most depleted values were in the Dmitry Laptev Strait, and the most enriched values were in the Offshore Arctic Shelf zone. The values from stations offshore the Indigirka River were more depleted than those offshore the Kolyma and Lena Rivers. The range of $\Delta^{14}\mathrm{C}_{OC}$ values is comparable to those measured by Vonk et al. (2012), and a comparison of the two datasets is shown in Figure S3.





## 3.2 Distributions of individual py-GCMS markers across the ESAS

Figure 2 shows the distribution of each compound group across the ESAS. The proportion of Phenols ranged from 3% to 62% (Figure 2A) with an average of 28% ±16% (1sd) . The value was highest in the Nearshore Laptev Sea (average 42%, n = 8) and DLS (average 49%, n = 2), and lowest in the far offshore parts of the Offshore Arctic Shelf zone (YS-88 and YS-100, both

3%). The proportion of Pyridines ranged from 8% to 74% (Figure 2B) with an average of 33% ±20% (1sd). The value was highest in the far offshore samples, YS-88 (74% ± 5%) and YS-100 (69% ±1%), while it is lowest in the Near Laptev Sea, at sample YS-15 (8% ±2%). Alkylbenzenes ranged from 1% to 10% (Figure 2C) with an average of 5% ±2% (1sd). The highest proportions were in the coastal areas near to the Dmitry Laptev Strait (average 9%), but not next to the Lena River mouth. The Kolyma River mouth sample, YS-34, also had high concentrations of Alkylbenzenes (9% ± 1%). Furfurals ranged from

6% to 38% (Figure 2D) with an average of 23% ±6% (1sd). Their distribution does not show a clear pattern across the ESAS, large proportions of Furfurals are found in both nearshore (TB-17, 37% ±6%) and offshore (YS-104, 37% ±3%) sediments. Aromatics ranged from 3% to 17% (Figure 2E) with an average of 9% ±4% (1sd). They were most concentrated in the area between the Yana and Indigirka Rivers, comprising the Dmitry Laptev Strait and the coastal area to the east of this (YS-22, YS-24, YS-26, YS-28; average 16%). Proportions were lowest in the far offshore samples (YS-88, YS-99, YS-100; average

4%). Cyclopentenones ranged from 0% to 3% (Figure 2F) with an average of 1% ±1% (1sd). Concentrations were highest in the western nearshore areas, close to the Lena River, in the Buor-Khaya Bay and in the Dmitry Laptev Strait. Proportions on the Offshore Arctic Shelf were negligible.

## 3.3 Regional variations in py-GCMS target compounds

Terrestrial ICD samples were dominated by Phenols, Furfurals and Pyridines, averaging 37%, 25% and 23% respectively.

Samples from the Kolyma River (code "CH") were richest in Phenols, up to 47%. There was not much difference between shallow, middle and deep samples. The Nearshore Laptev Sea samples were dominated by Phenols and Furfurals, averaging 42% and 23% respectively. Phenols proportions were highest close to the rapidly eroding Muostakh Island (YS-15, YS-17) and next to the Lena River mouth (TB-46) with proportions dropping further from the shoreline down to just 17% at site TB-17. Pyridines proportions were low, just 15% on average. Cyclopentenones proportions, at 1-3%, were the highest of any

region. The Lena River mouth samples (TB-46, TB-59) were highest in Cyclopentenones. The Dmitry Laptev Strait samples were dominated by Phenols (46% and 52%) but low in Pyridines (14% and 17%). The proportion of Aromatics was higher than average (13% and 16%), as was the proportion of Cyclopentenones (1% and 3%). Sample YS-24 also reported the lowest proportion of Furfurals of all samples (6%). The Nearshore East Siberian Sea region had a decreasing amount of Phenols in an offshore direction (51% down to 6%) and an increasing amount of Pyridines (16% up to 61%). The two samples away from the

river outflows (YS-26 and YS-31) were relatively low in Furfurals (15% and 18% respectively) but high in Alkylbenzenes (7% and 9%) and relatively high in Aromatics (16% and 15%). The Offshore Arctic Shelf region contained few Phenols (average 12%) and was dominated by Pyridines (average 55%). Other compounds that were relatively enriched closer to shore were also reduced here (Aromatics 6%, Alkylbenzenes 3%, Cyclopentenones 0-1%) but furfurals represent 24% of the material studied.





The area furthest offshore to the east of the sample area (YS-88 and YS-100) was the most dominated by Pyridines (74% and 69% respectively) and the most reduced in the other compounds (e.g. Phenols 3%, Aromatics 3%).

## 4 Discussion

### 4.1 Deposition of old carbon on the ESAS

The $^{14}C_{OC}$ results confirm observations that the ESAS sediments are dominated by old carbon (Vonk et al., 2012). The additional data collected in this study allows a comprehensive map of radiocarbon ages across the ESAS to be produced (Figure 4C). This shows that the oldest radiocarbon ages were measured in sediments from the Dmitry Laptev Strait region, while the youngest are found in the Offshore Arctic Shelf group, especially in the eastern East Siberian Sea. Even the youngest samples have quite negative $\Delta^{14}C_{OC}$ values, lower than -350 ‰. This has been interpreted as a large input of old carbon from ICD permafrost deposits, especially via coastal erosion. Very negative $\Delta^{14}C_{OC}$ values in the Dmitry Laptev Strait zone support this theory, since it is a region of high coastal erosion rates (see Figure 1, Lantuit et al., 2011), low fluvial input and low marine productivity (Sparkes et al., 2015).

### 4.2 Distribution of compounds along a river-shelf transect

To investigate offshore trends in the macromolecular groups, and to explore the relationships between them, sediments collected along a river-shelf transect from the outflow of the Kolyma River to the distal shelf were used as a case study. These sediments were: ICD sample CH (average values from top, middle and lower samples); YS-34; YS-35; YS-36; YS-37; YS-38; YS-39; YS-40; YS-41 and YS-90 (see Figure 1). All transect sediment samples were dominated by Furfurals, Phenols and Pyridines, which combined comprised 75 % (YS-37) to 90 % (YS-90) of the total abundance. In an off-shore direction, the relative Phenols abundance decreased from 50 % (at YS-34) to 11 % (at YS-90, Figure S2b). Phenols can have multiple terrestrial (lignin, tannins and proteins; van Bergen et al., 1998) or marine origins (algal polyphenols; van Heemst et al., 1999). However, a strong correlation with lignin concentrations, determined in the same sediments using the CuO oxidation method, ($r^2 = 0.98$, p < 0.01, n = 9; Figure S2B; Tesi et al., 2014) suggests that the Phenols in both the transect sediments and the ESAS as a whole ($r^2 = 0.67$, p < 0.01, n = 24; Figure 4A) are primarily lignin derived. According to Tesi et al. (2016), plant material in these sediments is associated with large particles that deposit rapidly nearshore; OC from these particles forms a very minor component of the offshore sediment. Phenols abundance was 11 % in the sediment collected at offshore station YS-90, and only 3% at nearby station YS-88, suggesting that terrestrial lignin-bearing particles were mostly deposited or degraded before they reached this part of the distal shelf. It also suggests that marine production of Phenols was minimal. There is a strong linear correlation between Phenols abundance and $\delta^{13}C_{OC}$ ($r^2 = 0.69$, p < 0.01, n = 9; Figure S2D), reinforcing the idea that the major source of Phenols was terrestrial plant material. However, the relationship is somewhat better represented as a logarithmic trend ($r^2 = 0.81$) in which relative abundance of Phenols diminishes faster than the bulk $\delta^{13}C_{OC}$ value in nearshore settings. This may be due to the higher concentration of lignin phenols in large particles which deposit closer to the shoreline



than the sediment as a whole. However, it has been shown that lignin phenols are present in all size fractions across the ESAS (Tesi et al., 2016), and therefore are not just tracking large terrOC-rich particles. The non-linear correlation with BIT values (Doğrul Selver et al., 2015), and the high abundance of Phenols within the ICD samples, suggests that this is not due to Phenols being dominantly river-derived material as has been suggested for branched GDGT biomarkers (Sparkes et al., 2015).

Pyridines abundance increased from 16 % to 64 % in an offshore direction along the same transect (Figure S2A) and dominated the sediment collected at station YS-90 (64 %). The increasing Pyridines abundance coincides with a shift towards more marine $\delta^{15}$N values ($r^2 = 0.75$, $p < 0.01$, $n = 9$; Figure S2C) and $\delta^{13}$C$_{OC}$ ($r^2 = 0.93$, $p < 0.01$, $n = 9$; Figure S2D). Although Pyridines were present in ICD samples (23%, comparable to the 16% found in the nearshore samples), these results suggest that in the ESAS sediments, particularly those further offshore, they were mainly of marine origin and the low-Phenols, high-

Pyridines pattern observed in the furthest offshore sediments could potentially be used as a marine endmember composition. Pyridines themselves are not a marker for marine OC since they are present in onshore samples, with plant proteins being a likely terrestrial source.

## 4.3 Distribution of phenols and pyridines across ESAS

The patterns seen along the Kolyma River-distal shelf transect suggest that the distribution of Phenols and Pyridines in sed-

iments may be usable as a proxy for measuring the relative input of terrestrial and marine carbon. There are several existing methods of performing this measurement, which can be used to test the applicability of the new pyrolysis-GCMS based approach. An index value, ranging from zero to one, can be obtained by comparing the relative peak areas of Phenols and Pyridines (NB: this index is not affected by changes in any other compounds):

$$\frac{Phenols}{Phenols + Pyridines} \tag{1}$$

In this index, a value of one means Phenols dominated, and is therefore interpreted as terrestrial in origin, and a value of zero means Pyridines dominated, interpreted as marine in origin. Our expansive dataset allows the Phenols-Pyridines ratio index (PPRI) to be examined as a proxy for terrOC across the entire ESAS. Figure 3 shows that the PPRI is highest in the Nearshore Laptev Sea (0.88), next to the coastline and the Lena River mouth, and is fairly high in all coastal settings west of the Kolyma River. This includes areas that have been described as river dominated and coastal erosion dominated in biomarker

studies (Bischoff et al., 2016; Sparkes et al., 2015). The value drops towards zero in distal offshore settings and east of the Kolyma River. In the western sections of the study area, offshore the Lena and Indigirka rivers, the transition from Phenols-rich to Pyridines-rich sediments occurs at about 200 km offshore. Offshore the Kolyma River this transition happens closer to shore. This pattern may be due to the enhanced export of lignin-derived phenol from the Lena River, due to its greater annual discharge (4.3 times more water, twice as much sediment; Gordeev, 2006) and less permafrosted basin area (71% continuous

permafrost in the Lena catchment compared to 99% for the Kolyma; Kotlyakov and Khromova, 2002). This would lead to a greater amount of terrestrial material (from the active layer at the top of the permafrost in both catchments, and also some deeper permafrost soil regions in the Lena catchment) being discharged from the Lena than the Kolyma. Alternatively, there





could be an increased proportion of Pyridines in the sediments offshore the Kolyma River due to the influx of marine OC-rich Pacific Ocean water through the Bering Strait (Semiletov et al., 2005; Bröder et al., 2016).

The PPRI can be compared to other terrestrial vs. marine proxy measurements in the region, namely the BIT index (Hopmans et al., 2004; Sparkes et al., 2015), $R'_{soil}$ proxy (Bischoff et al., 2016; Doğrul Selver et al., 2012) and $\delta^{13}C_{OC}$ (Vonk et al., 2012). Figure 5 shows the strong relationship between the Phenol-Pyridine Ratio and these alternative proxies. There is very strong positive correlation with $R'_{soil}$ ($r^2 = 0.80$, $p < 0.01$, $n = 38$) and very strong negative correlation with $\delta^{13}C_{OC}$ ($r^2 = 0.74$, $p < 0.01$, $n = 34$). There is also a significant correlation with BIT when analysed for a linear fit ($r^2 = 0.73$, $p < 0.01$, $n = 38$). However, Figure 5A shows that this relationship is likely non-linear in reality. The Nearshore Laptev Sea, Dmitry Laptev Strait and Nearshore East Siberian Sea samples show a fast reduction in BIT relative to PPRI, while Offshore Arctic Shelf samples have a range of PPRI values but uniformly low BIT. This relationship between BIT and PPRI is likely due to the biomarker sourcing and distribution patterns across the ESAS. Branched GDGT rich river sediment deposits rapidly close to river mouths, while Phenols, despite also depositing rapidly nearshore, are sourced from both rivers and coastlines, (see model in Sparkes et al., 2015). Whilst it must be recognised that the Py-GCMS method described here is limited to dealing only with relative proportions of each compound group, the strength of these correlations with other terrestrial-marine proxies suggests that the PPRI is a useful tool for measuring the source of macromolecular organic carbon in a sediment. As a relative measurement rather than reporting absolute values, the data cannot be used to interpret a relative rise or fall in a compound class concentration as indicative of enrichment or degradation in that particular class, rather as a change in the overall chemical composition of the sediment being examined. Hydrodynamic sorting of particulate matter, rather than selective production or degradation, may produce changes in relative concentration (Tesi et al., 2016). Despite these reservations, the Py-GCMS approach has the advantage that it samples the non-extractable portion of the organic matter, whereas biomarker studies concentrate on the smaller extractable portion. Further examination of the Phenol-Pyridine ratio, particularly in other laboratories or with other pyrolysis equipment, should be undertaken to test the widespread applicability of the ratio as a geochemical tool.

## 4.4 Furfurals and other compounds

Unlike Phenols and Pyridines, and despite their relative abundance ranging from 6% to 38%, there were no consistent nearshore-offshore patterns in the distribution of Furfurals across the ESAS (Figure 2D). One possible explanation for this observation is that Furfurals may be representing the pyrolysis products of carbohydrates from both terrestrial and marine organic matter. Previous studies have suggested that there is a transition from terrestrial to marine domination of OC across the shelf, both for bulk OC (Semiletov et al., 2005; Vonk et al., 2012) and molecular biomarkers (Karlsson et al., 2011; Sparkes et al., 2015; Vonk et al., 2010, 2014). Therefore, it is expected that common compounds present in both terrestrial and marine organic matter, such as carbohydrates, may not show a distinctive change in relative concentration from nearshore to offshore. Simple measurement of Furfurals may not show the transition from terrestrial to marine carbohydrate sources.

The across-shelf pattern of the minor compounds (Aromatics, Alkylbenzenes and Cyclopentenones) suggests that the sources of these may not be identical. For example, Aromatics are proportionally higher in regions far from the major river mouths (Dmitry Laptev Strait stations YS-22 and YS-24), areas which are dominated by coastal erosion (Vonk et al., 2012), and near





the Indigirka River (stations YS-26 and YS-28). In both of these areas, the proportion of Pyridines was low, but there was no correlation with other compound classes. Both Phenols and Cyclopentenones were high in the Dmitry Laptev Strait and low off the Indigirka River; Alkylbenzenes were high in the Dmitry Laptev Strait and at YS-26 but low at YS-28; Furfurals were low in the Dmitry Laptev Strait and high offshore the Indigirka River. This suggests that the delivery mechanism (i.e.

fluvial vs. coastal erosion) or offshore behaviour of Aromatics is unlike that of the other compound classes. A comparison with radiocarbon data (Vonk et al., 2012) suggests that Aromatics may be tracing the input of ancient terrOC to the ESAS. $^{14}C_{OC}$ data from this and previous studies is mapped across the ESAS in Figure 4C, and bears a striking resemblance to the distribution of Aromatics (Figure 2E). Figure 4B shows a negative correlation between relative proportion of Aromatics and $\Delta^{14}C_{OC}$ ($r^2$ = 0.44, p <0.01, n = 36). Note that there is weak or no correlation with other compound groups ($r^2$ values: Furfurals, 0.02;

Alkylbenzenes, 0.20; Phenols, 0.13; Cyclopentenones, 0.01; Pyridines, 0.18). ICDs are much older than fluvially eroded topsoil or marine productivity, with very little to no radiocarbon present ($^{14}C_{OC}$ = –940 ±84 ‰, n = 300; Vonk et al., 2012) and so areas dominated by coastal erosion rather than fluvial erosion have more negative $^{14}C_{OC}$ values. In the ESAS sediments this corresponds to the Dmitry Laptev Strait and Nearshore East Siberian Sea groups of samples. The youngest samples, with the least negative $^{14}C_{OC}$, are those from the Offshore Arctic Shelf group with the lowest proportion of Aromatics, thought to be

dominated by marine productivity. The BKB samples, highly influenced by river erosion of soils (Sparkes et al., 2015), also have relatively young radiocarbon ages and lower proportions of Aromatics. This correlation between older OC and aromatic compounds may be due to maturation of permafrost over time into simpler structures, especially into aromatic ones (Barden et al., 2011), and the protection of these compounds on mineral surfaces. Studies of soils from along an age gradient found that aromatic compounds were more likely to form mineral-organic associations, more resistant to biodegradation (Mikutta et al.,

2007, 2009). ICD samples were not enriched in Aromatics, so the source of these Aromatics is likely to be permafrost soil rather than ice complexes. The potential for pyrolysis-derived aromatic compounds being a tracer for very old permafrost material, especially coastal-erosion derived terrOC, should be investigated in future using more detailed sampling and compound-specific radiocarbon analysis. This would confirm the radiocarbon age of the aromatic compounds, as well as the mechanisms for their production and release.

**4.5  Principal Component Analysis shows offshore trends**

The changing proportions of each compound class, and a variety of proxy measurements (Bischoff et al., 2016; Sparkes et al., 2015; Vonk et al., 2012) were investigated by principal component analysis (PCA) using the software package "R". Principal components were calculated using the prcomp() function, which is a singular value decomposition method. Variables were automatically scaled and centred before analysis. Figure 6A shows the results of this analysis when performed on the py-

GCMS compound classes. Principal component 1, accounting for 58% of the variance, shows that the relative proportions of Alkylbenzenes, Aromatics, Phenols and Cyclopentenones are in opposition to the relative proportion of Pyridines. This pattern can be broadly seen in Figure 2, where the relative proportion of Pyridines is highest where the other four are lowest, and vice versa. This variance is interpreted as the difference between terrestrial and marine OC dominated sediments. A division of the PCA diagram at x = 0 shows that all Offshore Arctic Shelf samples lie on the Pyridines-dominated side of the chart,





and that all bar two of the Nearshore Laptev Sea, Dmitry Laptev Strait, Nearshore East Siberian Sea and ICD samples lie on the "terrestrial" side of the chart. Nearly orthogonal to these measurements is the proportion of Furfurals, which is the main variable of principal component 2 (18% of the variance). There are Offshore Arctic Shelf, Nearshore Laptev Sea and Nearshore East Siberian Sea sediments that are enriched in Furfurals. This shows that the relative concentration of Furfurals is not linked to the terrestrial – marine transition observed across the ESAS.

Figure 6B shows the results of principal component analysis carried out on the various terrestrial vs. marine proxies discussed in this paper (Phenol-Pyridine Ratio, BIT index, $R'_{soil}$ index, and $\delta^{13}C_{OC}$) and $\Delta^{14}C_{OC}$. Principal component 1, accounting for 80% of the variance, has $\delta^{13}C_{OC}$ in opposition to Phenol-Pyridine Ratio (PPRI), BIT and $R'_{soil}$. These four terrestrial *vs.* marine proxies are oriented opposite to $\delta^{13}C_{OC}$ in Figure 6B since the latter becomes more negative with increasing terrestrial material whereas the other proxies trend to higher values with increased terrOC. Therefore PC1 denotes the transition from terrestrial to marine dominance of the OC. It is notable that BIT lies slightly away from the PPRI, $R'_{soil}$ index and $\delta^{13}C_{OC}$ vectors. We interpret this as being due to the BIT index being strongly linked to river-derived terrOC in this region (Sparkes et al., 2015) whereas $\delta^{13}C_{OC}$ is measuring the entire sediment sample and $R'_{soil}$ is thought to measure both river and coastal derived terrOC (Bischoff et al., 2016). This offset is seen in the sample groups – the samples from the Nearshore Laptev Sea, especially those near to the Lena River, plot close to the BIT vector whilst the coastal erosion dominated Dmitry Laptev Strait sample, along with the Nearshore East Siberian Sea samples, plot further from the BIT vector. The $\Delta^{14}C_{OC}$ vector is at 45° to the trend in terrestrial-marine proxies. This is interpreted as showing that all marine OC is dominated by young OC (high $\Delta^{14}C_{OC}$ values), as is material coming from the rivers. Coastal erosion sediments contain older material and therefore plot opposite to the $\Delta^{14}C_{OC}$ vector.

Therefore we can define three areas on the PCA diagram (Figure 6B). As mentioned, PC1 divides the diagram into terrestrial and marine sections. Within the terrestrial half of the diagram, PC2 differentiates between river derived and coastal erosion derived terrOC. Following the offshore trends of each major river (shown in Figure 6B), there is a transition from river-influenced terrOC to ICD-influenced terrOC and finally marine OC dominated compositions. The Lena River is the most river-influenced trend, followed by the Kolyma River, with the Indigirka River offshore transect mostly dominated by terrOC from ice complexes. The Indigirka River sits between two areas of extremely high coastal erosion rates (Lantuit et al., 2011), so this is not unexpected. These patterns agree with the model published in Sparkes et al. (2015), which predicted a transition from river derived upper permafrost to coastal erosion sourced ICD material to marine OC with distance offshore. Overall, principal component analysis has proven to be a valuable tool for understanding the transition between OC types across the ESAS. Multiple organic proxies agree that there is a large amount of terrestrial OC on the ESAS, and that erosion of coastal sediments greatly increases the delivery and burial of terrestrial OC if compared to purely river-derived material.

## 5 Conclusions

By analysing sediment samples from across the East Siberian Arctic Shelf using the relatively rapid py-GCMS technique and categorising major pyrolysis moieties into a number of source-related classes, clear offshore trends were observed. Analyses



indicated that nearshore samples were rich in Phenols, Aromatics, Alkylbenzenes and Cyclopentenones, which all decreased in importance offshore, suggesting a terrestrial source. Relative abundance of Pyridines increased offshore, suggesting a marine source, whilst Furfurals were present everywhere and may have been sourced from both terrestrial and marine carbohydrates. We propose that comparing the relative abundance of Phenols to the sum of Phenols and Pyridines (Phenol-Pyridine Ratio

Index; PPRI) is a novel, useful tool for estimating the input of terrestrial and marine macromolecular OC in offshore sediments. Both a subsample set from the Kolyma River as well as sediments from across the entire ESAS, show, for the first time, a strong correlation between the py-GCMS results (both relative values and the Phenol-Pyridine Ratio) and previous, independent measurements of offshore terrOC (BIT index, $R'_{soil}$ index, lignin phenols). Principal component analysis, carried out on a large number of different measurements performed on these sediments, showed the offshore trend from river and coastal

erosion derived material to marine OC across the ESAS and demonstrates the value of a holistic, multi-proxy approach to understanding the carbon cycle in complex environments.

*Author contributions.* Ö. Gustafsson, B. E. van Dongen, and I. P. Semiletov collected samples along with the crew of ISSS-08. H. M. Talbot and B. E. van Dongen designed the study. Py-GCMS measurements were carried out by R. B. Sparkes and A. Doğrul Selver. $\Delta^{14}$C measurements were carried out by N. Haghipour, L. Wacker and T. I. Eglinton. R. B. Sparkes, A. Doğrul Selver, H. M. Talbot, and B. E. van

Dongen prepared the manuscript with contributions from all co-authors.

## Data Availability

Data presented in this manuscript is included in the supplementary file (Tables S2, S3). Raw radiocarbon data is availble as Table S4.

*Acknowledgements.* We gratefully acknowledge receipt of a NERC research Grant (NE/I024798/1 and NE/I027967/1) to B. E. van Dongen and H. M. Talbot., a Ph.D. studentship to A. Doğrul Selver funded by the Ministry of National Education of Turkey, and support from the

Government of the Russian Federation (mega-grant 14.Z50.31.0012) to I. Semiletov. We thank the crew and personnel of the R/V Yakob Smirnitskyi and all colleagues in the International Siberian Shelf Study (ISSS) Program for support, including sampling. We thank T. Tesi for providing the Yedoma samples for the Kolyma and Indigirka catchment areas. The ISSS program is supported by the Knut and Alice Wallenberg Foundation, the Far Eastern Branch of the Russian Academy of Sciences, the Swedish Research Council (VR Contract No. 621-2004-4039, 621-2007-4631 and 621-2013-5297), the US National Oceanic and Atmospheric Administration (OAR Climate Program Office,

NA08OAR4600758/Siberian Shelf Study), the Russian Foundation of Basic Research (08-05-13572, 08-05-00191-a, and 07-05-00050a), the Swedish Polar Research Secretariat, the Nordic Council of Ministers and the US National Science Foundation (OPP ARC 0909546). Finally, we thank the associate editor and XXXXXX reviewers for constructive suggestions.



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





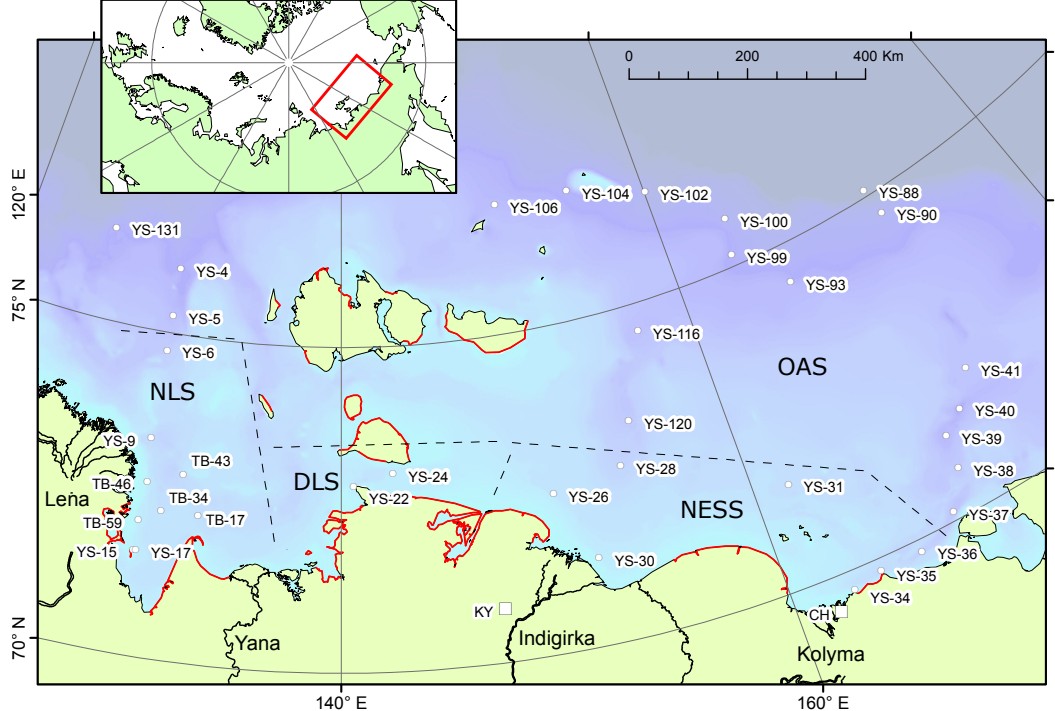

**Figure 1.** Map of the East Siberian Arctic Shelf (ESAS) showing the location of surface sediment samples (white circles) and ice complex samples (white squares) used in this study. Areas of rapid coastal erosion (>1m y-1; Lantuit et al., 2011) are shown in red. Regions of the ESAS referred to in the paper are shown using dashed lines (DLS = Dmitry Laptev Strait, NLS = Nearshore Laptev Sea, NESS = Nearshore East Siberian Sea, OAS = Offshore Arctic Shelf).

Yunker, M. B., Macdonald, R. W., Veltkamp, D. J., and Cretney, W. J.: Terrestrial and marine biomarkers in a seasonally ice-covered Arctic estuary – integration of multivariate and biomarker approaches, Marine Chemistry, 49, 1–50, doi:10.1016/0304-4203(94)00057-K, doi: DOI: 10.1016/0304-4203(94)00057-K, 1995.

Zimov, S. A., Davydov, S. P., Zimova, G. M., Davydova, A. I., Schuur, E. A. G., Dutta, K., and Chapin, F. S.: Permafrost carbon: Stock and decomposability of a globally significant carbon pool, Geophysical Research Letters, 33, L20 502, doi:10.1029/2006GL027484, 2006.





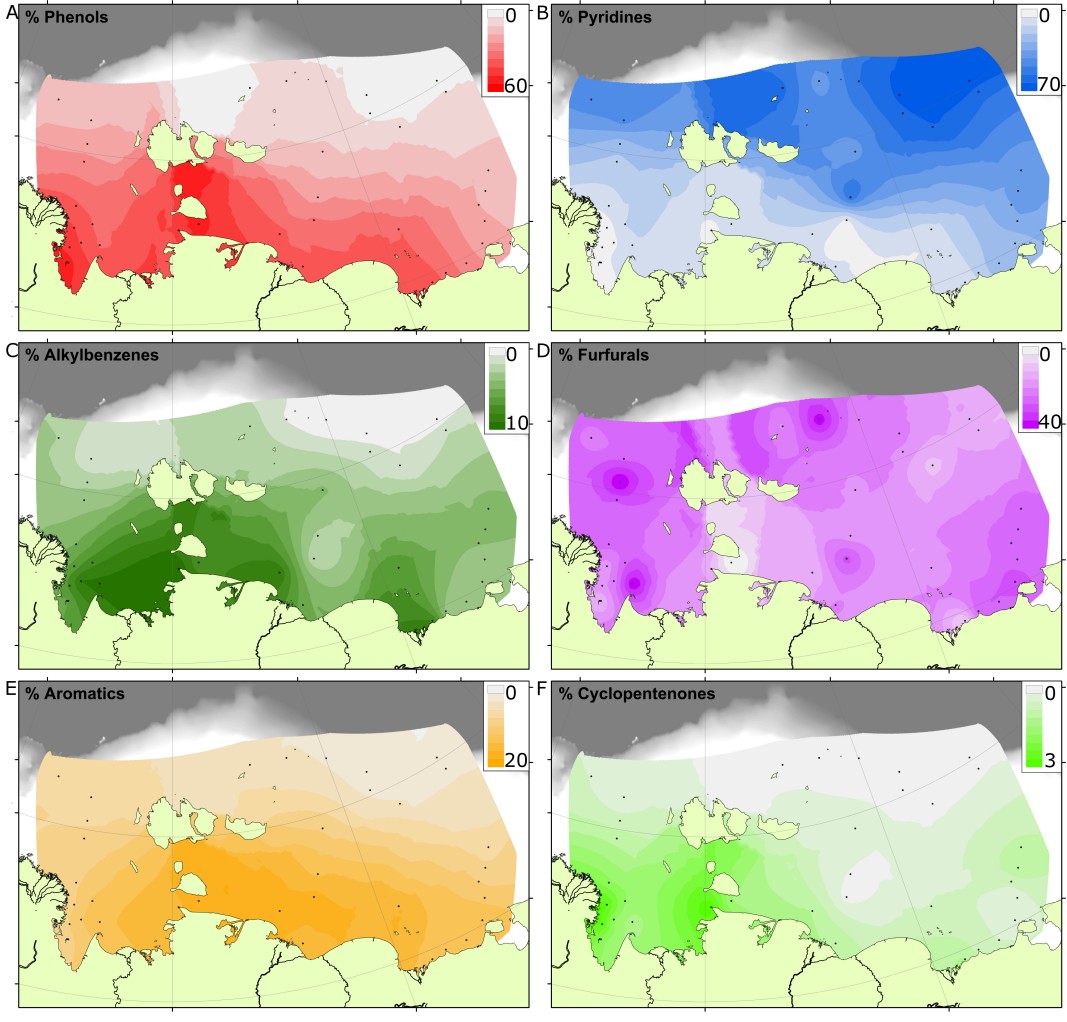

**Figure 2.** Distribution of (pyrolysis) compound classes across the ESAS. Distributions are reported as "% of total", comparing the peak area of the major ion(s) in the compound class to the total peak area of major ions of all compound classes. See Table S2 for the breakdown of relative areas for each measured compound. Distributions are reported as a colour gradient from full-colour (maximum observed) to white (zero). Sample locations are shown as black dots. Interpolation between sample sites was carried out using a 'kriging' algorithm within software package *ArcGIS*.



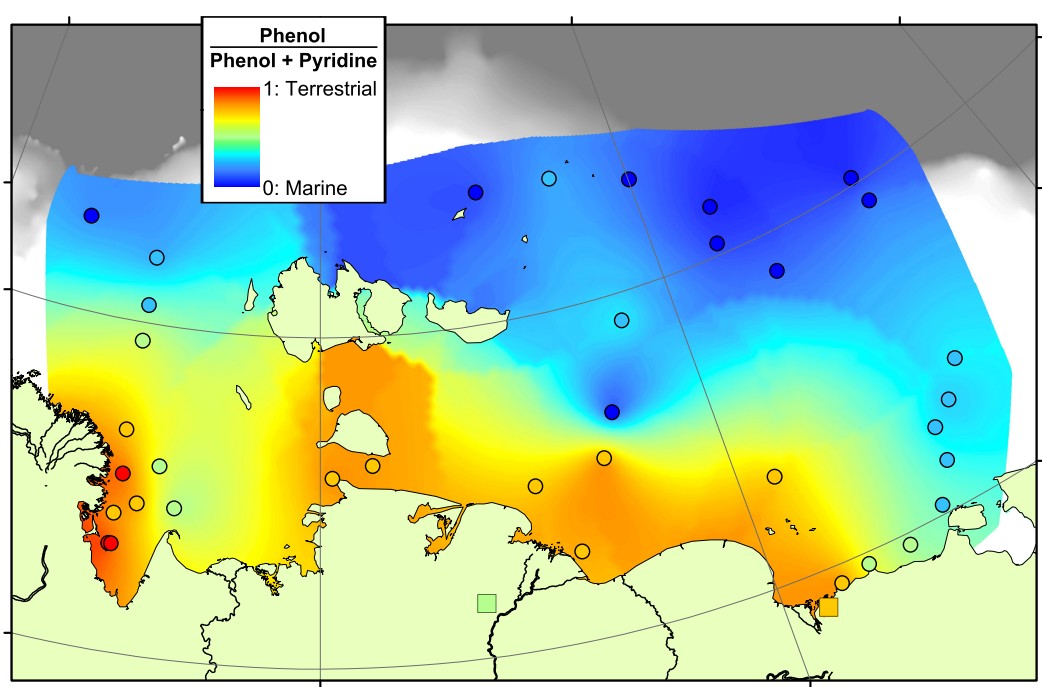

**Figure 3.** Map of the phenol-pyridine ratio across the ESAS (Ratio is calculated from the relative abundances of Phenol / (Phenol + Pyridine); higher values are interpreted as being terrestrial-dominated). Coloured circles show the ratio measured in each offshore sample, squares show onshore ice complex sample values. Interpolation between samples was carried out using a 'kriging' algorithm within software package *ArcGIS*.



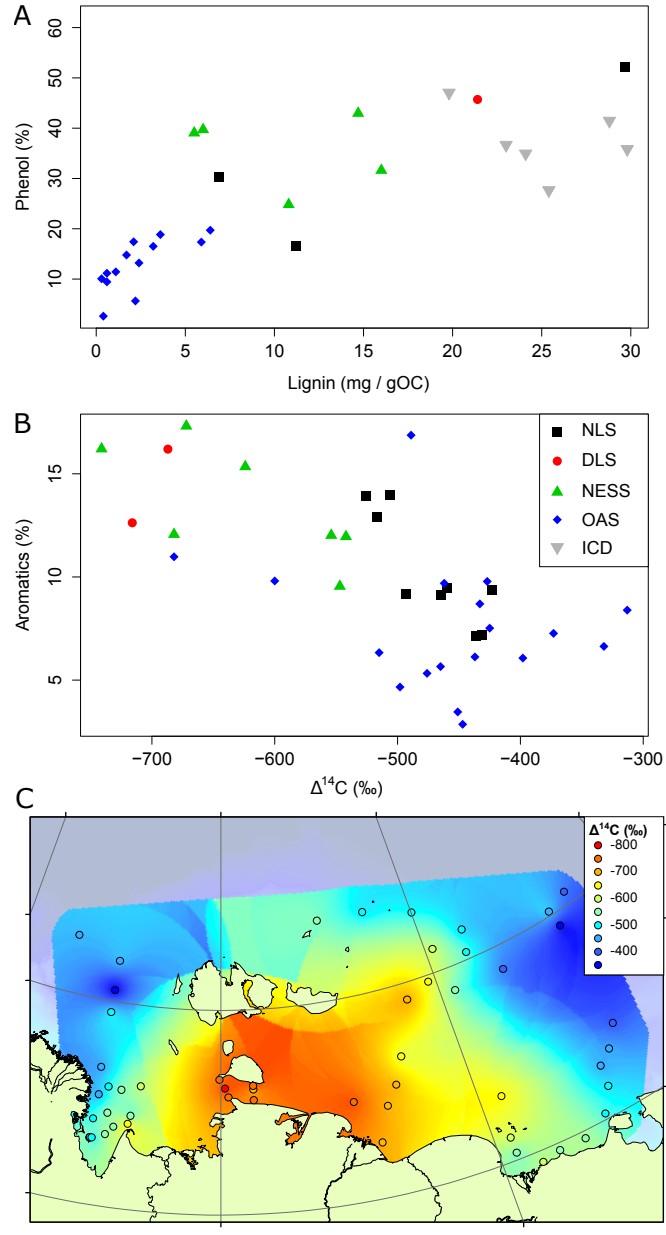

**Figure 4.** Correlation plots of A) Relative abundance of phenols vs. measured concentration of lignin phenols in identical samples as measured by Tesi et al (2014) and B) Relative abundance of aromatics vs. $\Delta^{14}C_{OC}$ measurements from this study and Vonk et al. (2012). In each case, samples are distinguished by sample area (see legend; sample areas defined in Figure 1). C) Map of radiocarbon ($\Delta^{14}C_{OC}$) values measured in this study and Vonk et al. (2012). Interpolation of $\Delta^{14}C_{OC}$ data was performed using a 'kriging' algorithm within software package *ArcGIS*. (DLS = Dmitry Laptev Strait, NLS = Nearshore Laptev Sea, NESS = Nearshore East Siberian Sea, OAS = Offshore Arctic Shelf)





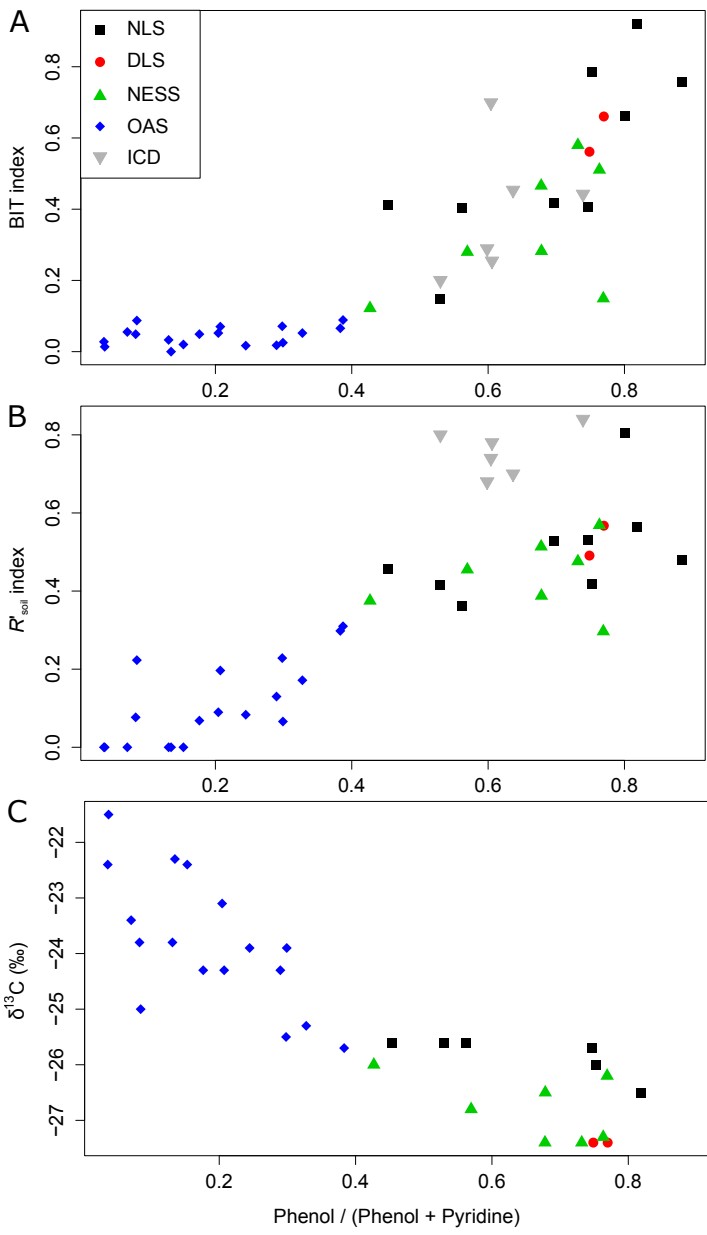

**Figure 5.** The Phenol-Pyridine Ratio plotted against A) BIT, B) $R'_{soil}$ and C) $\delta^{13}C_{OC}$. In each case there is a strong correlation between the proxies. (DLS = Dmitry Laptev Strait, NLS = Nearshore Laptev Sea, NESS = Nearshore East Siberian Sea, OAS = Offshore Arctic Shelf)





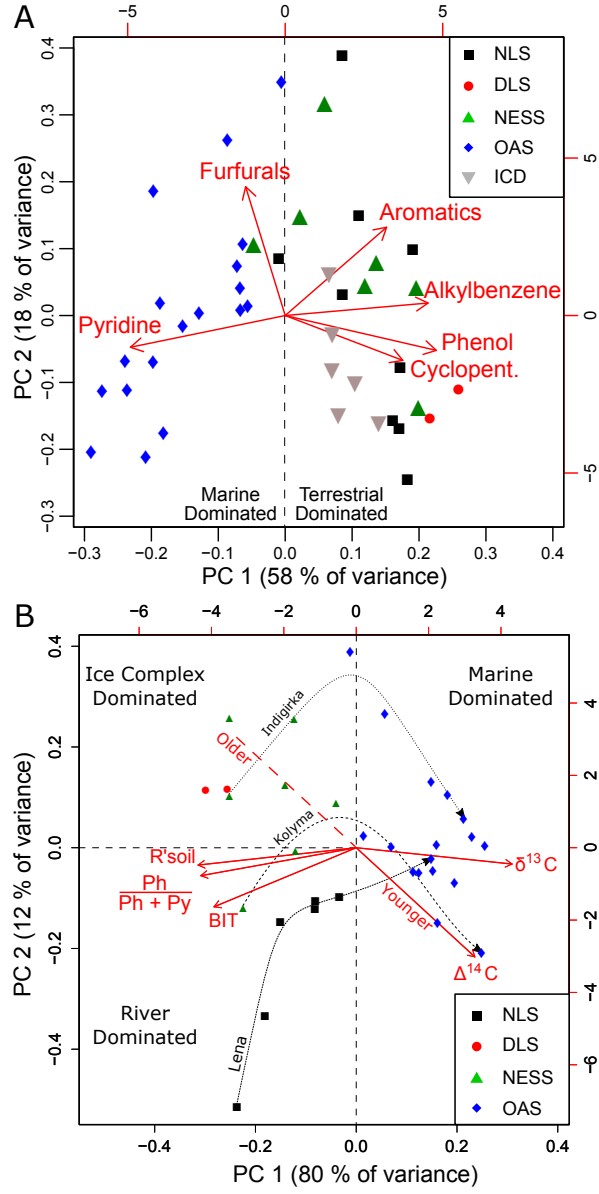

**Figure 6.** Principal component analyses of A) various measured compounds from py-GCMS analysis (see Figure 2 and Table S2) and B) various terrestrial-marine proxies. Sample location regions are represented by symbol shapes and colours (see legend). Inferred domains of marine and terrestrial (split into river and ICD sections in panel B) dominance are shown with straight dashed lines. Offshore transects of surface sediments from major rivers to the ESAS (panel B) are shown using curved dotted lines and labelled with the river name at the nearshore end of the offshore transect. (DLS = Dmitry Laptev Strait, NLS = Nearshore Laptev Sea, NESS = Nearshore East Siberian Sea, OAS = Offshore Arctic Shelf)





**Table 1.** Representative moieties analysed in this study, and the compound groups that they are interpreted to represent (after Guo et al., 2004).

| Compound group | Compound name | Major ion $M_w$ | Class represented |
|---|---|---|---|
| Phenols | Phenol | 94 | Lignin |
| Pyridines | Pyridine | 79 | Marine N-rich |
| | Methyl Pyridine | 93 | primary productivity |
| Alkylbenzene | Dimethyl Benzene | 106 | Anaerobic soils |
| Furfurals | Furfural | 96 | Less-degraded carbohydrates, |
| | Methyl Furfural | 110 | both marine and terrestrial |
| Aromatics | Indene | 116 | Mature OC |
| | Napthalene | 128 | |
| Cyclopentenones | Methylcyclopentenone | 96 | Soil polysaccharides |