# Peer review of "Macromolecular composition of terrestrial and marine organic matter in sediments across the East Siberian Arctic Shelf"

_The Cryosphere, 2016_

## Referee Comment (RC1) · Anonymous Referee #1 · 24 Aug 2016

Review of Cryosphere manuscript (MS #: doi:10.5194/tc-2016-143, 2016) Title: Macro-molecular composition of terrestrial and marine organic matter in sediments across the East Siberian Arctic Shelf Authors: Sparkes et al.

The manuscript by Sparkes et al. set out to investigate macromolecular composition of terrestrial and marine organic matter in sediments across the East Siberian Arctic Shelf using pyrolysis-GC-MS and radiocarbon measurements. They found that there is a strong offshore trend from terrestrial phenols, aromatics, cyclopentenones to marine pyridines, but not for furfurals which represent carbohydrate component. They also found a good agreement between phenols measured using py-GCMS and independent quantification of lignin phenol. Combined with radiocarbon composition of the bulk OC,

the authors suggested that the aromatics components were derived from old terr-OC while the phenols groups were sourced from modern woody materials. Based on their new data, the authors proposed a compelling index, using the relative proportions of phenols and pyridines, to be used as a novel proxy for terrestrial vs. marine organic components in the East Siberian Arctic Shelf sediment.

In general, this manuscript presents a great quantitative data set based on molecular characterization using pyrolysis-GC-MS and radiocarbon analyses of bulk OC. The index of the proportions of phenols and pyridines seems to be working in identifying the relative contribution of macromolecular organic matter in this complex coastal marine environment in the Arctic Ocean. Results shown in Figure 5 are quite convincing and support the authors' proposed index. I support the publication of this work.

Specific comments: a) Section 2.1: Materials in this section, although important, can be tied up or shortened a bit. b) Throughout the manuscript, Why names of organic compound classes, such as Phenols, Aromatics, Alkylbenzenes and Cyclopentenones, etc., need capitalized for the first character? c) Pg-2, Line-14: Add reference of Ping et al (2011) for pan arctic coastal erosion: Ping, C.-L., G.J. Michaelson, L. Guo, M.T. Jorgenson, M. Kanevskiy, Y. Shur, F. Dou and J. Liang. 2011. Soil carbon and material flux across the eroding coastline of the Beaufort Sea, Alaska. JGR-Biogeosciences, 116, G02004, doi:10.1029/2010JG001588 d) Section 3.1: change " Bulk radiocarbon measurements" to read: " Bulk radiocarbon composition" e) Section 4.5: Regarding title of this section, instead of a sentence, I suggest the use of a regular title.

Overall, I support publication of this manuscript.

---

## Referee Comment (RC2) · Anonymous Referee #2 · 11 Sep 2016

Comments on Manuscript tc-2016-143,

Macromolecular composition of terrestrial and marine organic matter in sediments across the East Siberian Arctic Shelf

by R. B. Sparkes et al.

This is an interesting and well written paper that fits perfectly into the scope of the inter-journal Special issue " Climate–carbon–cryosphere interactions in the East Siberian Arctic Ocean: past, present and future" in "The Cryosphere".

The work is based on an analysis of more than 40 sediment samples from onshore (6) and off-shore (36) sites across the East Siberian Arctic Shelf (ESAS), acquired during

the ISSS-08 expedition. The authors use pyrolysis gas chromatography mass spectrometry (py-GCMS) to investigate, across the ESAS, all major groups of macromolecular composition of terrestrial and marine organic matter in sediments, rather than only the solvent ones. Nine compound groups are identified and linked to particular groups of terrestrial and marine macromolecular materials. Based on the abundance of Phenols and Pyridins in sediment samples along a river-shelf transect, the authors propose a novel index (the PPRI index, Phenol-Pyridine Ratio Index) for estimating the input of terrestrial (dominant: phenol) and marine (dominant: pyridin) macromolecular organic carbon in offshore sediments. In addition, minor compound groups occurring in samples across the ESAS are analysed and their spatial distribution is discussed. Principal Component Analysis is shown to be a valuable tool for understanding the pattern of transition from terrestrial organic carbon to marine organic carbon dominated compositions across the ESAS. The data, analysis and conclusions presented in the manuscript are convincing, and I recommend publication of the manuscript.

---

## Author Comment (AC1) · 22 Sep 2016

We thank the reviewer for their endorsement of the paper, and appreciate their recommendation to publish the manuscript without modifications.

---

## Author Comment (AC2) · 22 Sep 2016

We thank the reviewer for their support of this work, and answer their comments below:

a) Section 2.1: Materials in this section, although important, can be tied up or shortened a bit.

We will carefully revise this section and remove any unnecessary information.

b) Throughout the manuscript, Why names of organic compound classes, such as Phenols, Aromatics, Alkylbenzenes and Cyclopentenones, etc., need capitalized for the first character?

We prefer to keep the capitalisation, because these classes represent a large group of molecules than the specific, representative compounds that were measured in this study. For example "Phenols" corresponds to one particular compound in the chromatogram which is being used as the representative compound, but "phenols" corresponds to the entire group of phenol compounds. However, if the editor prefers a change to the lower case we will happily make this alteration.

c) Pg-2, Line-14: Add reference of Ping et al (2011) for pan arctic coastal erosion: Ping, C.-L., G.J. Michaelson, L. Guo, M.T. Jorgenson, M. Kanevskiy, Y. Shur, F. Dou and J. Liang. 2011. Soil carbon and material flux across the eroding coastline of the Beaufort Sea, Alaska. JGR-Biogeosciences, 116, G02004, doi:10.1029/2010JG001588

We will add this reference at the identified location.

d) Section 3.1: change "Bulk radiocarbon measurements" to read: "Bulk radiocarbon composition"

We will change this as requested.

e) Section 4.5: Regarding title of this section, instead of a sentence, I suggest the use of a regular title.

We will change the section title to "Principal Component Analysis".

---

## Author Response (AR1)

We thank the reviewer for their support of this work, and answer their comments below:

a) Section 2.1: Materials in this section, although important, can be tied up or shortened a bit.

We have carefully revised this section, making a more concise introduction to the study area. Changes are shown in the marked-up manuscript.

b) Throughout the manuscript, Why names of organic compound classes, such as Phenols, Aromatics, Alkylbenzenes and Cyclopentenones, etc., need capitalized for the first character?

We prefer to keep the capitalisation, because these classes represent a large group of molecules than the specific, representative compounds that were measured in this study. For example "Phenols" corresponds to one particular compound in the chromatogram which is being used as the representative compound, but "phenols" corresponds to the entire group of phenol compounds. However, if the editor prefers a change to the lower case we will happily make this alteration.

c) Pg-2, Line-14: Add reference of Ping et al (2011) for pan arctic coastal erosion: Ping, C.-L., G.J. Michaelson, L. Guo, M.T. Jorgenson, M. Kanevskiy, Y. Shur, F. Dou and J. Liang. 2011. Soil carbon and material flux across the eroding coastline of the Beaufort Sea, Alaska. JGR-Biogeosciences, 116, G02004, doi:10.1029/2010JG001588

We have added this reference at the identified location.

d) Section 3.1: change "Bulk radiocarbon measurements" to read: "Bulk radiocarbon composition"

We have changed this as requested.

e) Section 4.5: Regarding title of this section, instead of a sentence, I suggest the use of a regular title.

We have changed the section title to "Principal Component Analysis".

[revised manuscript text omitted]